# Large Field Alpha Irradiation Setup for Radiobiological Experiments

**DOI:** 10.3390/mps2030075

**Published:** 2019-08-28

**Authors:** Stefan J. Roobol, Jasper J.M. Kouwenberg, Antonia G. Denkova, Roland Kanaar, Jeroen Essers

**Affiliations:** 1Department of Molecular Genetics, Erasmus University Medical Center, 3000 CA Rotterdam, The Netherlands; 2Oncode Institute, Erasmus University Medical Center, 3000 CA Rotterdam, The Netherlands; 3Department of Radiology & Nuclear Medicine, Erasmus University Medical Center, 3000 CA Rotterdam, The Netherlands; 4Department of Radiotherapy, Erasmus University Medical Center, 3015 GD Rotterdam, The Netherlands; 5Department of Radiation Science and Technology, Delft University of Technology, 2629 JB Delft, The Netherlands; 6Department of Radiation Oncology, Erasmus University Medical Center, 3000 CA Rotterdam, The Netherlands; 7Department of Vascular Surgery, Erasmus University Medical Center, 3000 CA Rotterdam, The Netherlands

**Keywords:** alpha particle, irradiation, microscopy, DSB, 53BP1, DNA damage, FNTD, dosimetry, external irradiation, clonogenic survival

## Abstract

The use of alpha particles irradiation in clinical practice has gained interest in the past years, for example with the advance of radionuclide therapy. The lack of affordable and easily accessible irradiation systems to study the cell biological impact of alpha particles hampers broad investigation. Here we present a novel alpha particle irradiation set-up for uniform irradiation of cell cultures. By combining a small alpha emitting source and a computer-directed movement stage, we established a new alpha particle irradiation method allowing more advanced biological assays, including large-field local alpha particle irradiation and cell survival assays. In addition, this protocol uses cell culture on glass cover-slips which allows more advanced microscopy, such as super-resolution imaging, for in-depth analysis of the DNA damage caused by alpha particles. This novel irradiation set-up provides the possibility to perform reproducible, uniform and directed alpha particle irradiation to investigate the impact of alpha radiation on the cellular level.

## 1. Introduction

Understanding the impact of alpha particles on biological material, such as DNA, is of utmost importance to verify and optimize future radionuclide therapy. Current studies in radiobiology focus on different radiation types with respect to the biological harm that different isotopes can induce in cellular systems. However, the exact biological effects of alpha particles, in the context of DNA damage, is still poorly understood. The high linear energy transfer (LET) of alpha particles, compared to beta- and gamma- irradiation, induces more cell death, which results in high relative biological effectiveness (RBE) [1]. This effectiveness is due to the short distance between individual ionization caused by alpha particles [2,3]. In addition, the highly ionizing path of alpha particles induces clusters of double stranded breaks (DSBs) in the DNA along a straight track (10−20 DSBs per 10 μm track length) [4]. Therefore, the use of alpha particle emitting radionuclides, conjugated to antibodies or peptides shows great promise improving radiotherapy in the clinic, through specific targeting and by exploiting the short path length to limit damage of healthy tissue [5,6,7,8].

The development of experimental alpha particle irradiation has seen a lot of attention in the past [9,10,11,12,13,14,15,16,17]. These studies mainly focused on dosimetry and bystander effects. More recent setups show increased complexity in radiation procedures but great promise in experimental radiobiology [18,19,20,21,22]. However, the active surface of commercially available alpha particle sources (e.g., from Eckert and Ziegler) are often smaller in diameter than culture dishes, thereby precluding quantitative cell colony formation assays, and require optimized protocols for alpha-track irradiation [22,23]. To allow an uniform alpha irradiation of large fields of cells using a small source a novel irradiation setup was developed [24]. Here we describe the procedure for large field irradiation using a relatively small alpha particle emitting ^241^Am source. In addition, this procedure was used for a novel method for alpha radiation (micro-) dosimetry using fluorescent nuclear track detectors (FNTDs) [16,24,25]. Using computer directed irradiation on cell populations cultured in Mylar dishes allows elaborate cell population assays, compared to previously reported methods [4,9,10,11,12,13,14,15,17,18,19,20,21,22,23,26,27]. Moreover, the irradiation procedure has been adjusted for cell culture on glass-coverslips, allowing to avoid the Mylar-based culture conditions and the possibility for super-resolution imaging. With the use of glass coverslips, repair proteins of several DSB-repair pathways can be studied in high resolution using techniques such as stochastic optical reconstruction microscopy (STORM) or structured illumination microscopy (SIM) [28]. With the use of the right materials and conditions this protocol could yield fast and reliable answers to biological questions regarding alpha particle induced DSBs or cell survival after irradiation with alpha particles. The setup was validated using immunofluorescence in combination with the use of SIM as super resolution technique and clonogenic survival assays were carried out, which demonstrate the effective irradiation of larger areas of cells.

## 2. Experimental Design

This protocol was developed in need of biological assays using alpha particle irradiation. In this study we used U2OS cells as model cell culture and irradiated the cells using alpha particles emitted by a ^241^Am source. The active surface of the source was controlled by an in house built automated stage for precise dosimetry during irradiation (Figure 1). By culturing cells on Mylar, a very thin foil, the alpha particles were able to reach the cells, allowing the cells to be irradiated from beneath whilst in their normal culture medium. In addition, this protocol describes a procedure to irradiate U2OS cells cultured on glass coverslips by alpha particles. This technique allowed super resolution imaging on irradiated cells.

This protocol is divided in several steps required to achieve the goals stated above. First, the preparation of custom-made Mylar dishes. Second, culturing U2OS cells using Mylar dishes or glass coverslips. Third, the irradiation procedure for both a large field of cells or only a specific area on the coverslip. Irradiated cells can be used for conventional clonogenic survival assays or immunohistochemistry.

### 2.1. Materials

Acetone (Sigma-Aldrich, Zwijndrecht, The Netherlands, Cat. No. 154598-1L)Ethanol absolute (VWR Chemicals, Paris, France, Cat. No. 83813.360)Sterile Distilled WaterDulbecco’s Phosphate-buffered saline, pH 7.4 (PBS, Sigma-Aldrich, Cat. No. D8537-500ML)Dulbecco’s Modified Eagle Medium (DMEM, Gibco, Thermo Fisher, Waltham, MA, USA, Cat. No. 11965092)Fetal Bovine Serum (Capricorn Scientific, Ebsdorfergrund, Germany, Cat. No. FBS-12A)Penicillin/streptomycin (Gibco, Thermo Fisher, Cat. No. 15140122)0.05% (w/v) Trypsin 0.53 EDTA solution (Sigma-Aldrich, Cat. No. T3924-500MLVectashield with DAPI (Vector Laboratories, Burlingame, CA, USA, Cat. No. H-1200)HEPES Buffer (Lonza, Portsmouth, NH, USA, Cat. No. 17-737E, pH: 6.98 – 7.3, Counter ion: NaCl)

### 2.2. Equipment

Pipette 1 mL (Gilson, Den Haag, The Netherlands, Cat. No. F123602)Laboratory tweezers (Fine Science Tools, Heidelberg, Germany, Cat. No. 11252-00)Alpha particle emitting ^241^Am source of 1.1 cm in diameter with an activity of 409.6 kBq (Czech Metrological Institute, Jihlava, The Czech Republic)Polyster plain film (Mylar), 1.9 μm thickness (Birkelbach Kondensatortechnik, Erndtebrueck, Germany, Cat. No. PTPL)Cardboard (any type will suffice, dimensions at least: 50 × 50 cm)Adhesive tape (any type will suffice)Lumox 35 dishes (Sarstedt, Nümbrecht, Germany, Cat. No. 94.6077.333)10 cm culture dishes (Greiner Bio-One, Alphen aan den Rijn, The Netherlands, Cat. No. 664160)Sandpaper (any type will suffice)Sanyo MCO-17AIC Copper Alloy CO_2_ Incubator (Marshall Scientific, Hampton, NH, United States, Cat. No. SANYOMCO)Microscope slide coverslips with 10 mm ø or smaller (Thermo Fisher Scientific, Cat. No. CB00100RA020MNT0)Disposable Scalpels - Sterile (Swann-Morton, Sheffield, United Kingdom, Cat. No. 0503)UV-irradiation set-up (Erasmus MC, Rotterdam, The Netherlands, Homemade)2× Motorized linear stages, resolution: 0.625 μm (Optics-Focus, Beijing, China, Cat. No. MOX-02-50)Manual laboratory jack, resolution: 0.01 mm (Optics-Focus, Cat. No. 8MAJ60)Motion Controller (Optics-Focus, Cat. No. MOC-01-1-220)Metal frame (TU Delft, Delft, The Netherlands, Homemade, Appendix A)3-D printed Dish holder of ABS plastic(TU Delft, Homemade, Appendix A)3-D printed Collimator of ABS plastic (TU Delft, Homemade, Appendix A)Large Polymethyl Methacrylate box to case the whole set-up and for heating, volume of 0.22 m^3^ (TU Delft, Homemade)Thin washers (Modelfixings.co.uk, Nottingham, United Kingdom, Cat. No. MF-SHW71301)Computer running MatLab (MathWorks, Natick, MA, USA)Compact Hairdryer (BaByliss, Paris, France, Cat. No. D212E)ThermoControl 2 (Lucky Reptile, Waldkirch, Germany, Cat. No. LR62121)

## 3. Procedure

### 3.1. Calibration of the Irradiation Stage (Recalibrate This Once a Month). Time for Completion: 30 min

Bolt the culture dish holder and the collimator on the aluminum frame (Figure 1B, Appendix A).Align the centers of the collimator and the dish holder. The alignment can be judged visually when the collimator is raised to the level of the culture dish holder. Store the x, y-coordinates of the linear stages (Figure 1A) in the control software after alignment.Adjust the gap between upper rim (e.g., the bottom of the culture dish, Figure 1C) and the top of the collimator to the appropriate size (2.8 mm) using the laboratory jack and a caliper (Figure 1D). Measure the gap size near each leg of the culture dish holder to check for a possible tilt. Remove this tilt by raising the legs using thin non-corroding washers so that the measured gaps near the legs are equal.
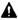
**CRITICAL STEP:** When the dish holder has not been removed, steps 2 and 3 can be skipped. A monthly check of the gap size and culture dish holder tilt is in this case sufficient.Place the circular 241Am source in the holder with collimator and bolt the assembly on the laboratory jack on the linear stages (Figure 2).Cover the whole set-up with the large Perspex box and place the Compact Hairdryer inside connected to the ThermoControl 2 for temperature control. Set the ThermoControl 2 to 37 °C.

### 3.2. Mylar Dish Preparation. Time for Completion: 3 h

Stretch the Mylar on the piece of cardboard, fix edges using tape. Make sure there are no apparent damages on the Mylar surface before use.Remove the factory applied foil and scour the bottom of the Lumox dishes using sandpaper. Wash both the scoured dish and lid of the Lumox dish and dry before the next step.Pour a small volume of acetone into a glass dish. Dip the scoured and cleaned Lumox ring in the acetone and keep in place for 10 s. Remove the ring from the acetone and move over to the stretched Mylar. Press the ring firmly on the, undamaged, Mylar and hold for 30 s. Let dry for at least 2 h.
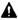
**CRITICAL STEP**: Do not move the ring while pressing on the Mylar, this could cause leakage later on. Make sure the Mylar surface inside the Lumox ring is damage free.Add 1 mL 70% ethanol to the dish to check for possible leaking spots. Aspirate the Ethanol if the Mylar dish is not leaking.Carefully cut the Mylar around the Mylar dish (minimum of 0.5 cm away from the edge) and store the Mylar dishes in 10 cm tissue culture dishes (2 per 10 cm dish).Sterilization of Mylar Dishes can be done by UV (~140 J/m^2^) irradiation or 70% ethanol wash (minimum 3 times). Always wash with sterile PBS or sterile MQ water.
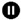
**PAUSE STEP**: The protocol can be paused here for a maximum of 3 days. Store the Mylar dishes at 4 °C and keep moist with PBS.

### 3.3. Cell Culture. Time for Completion: 30 min

Depending on the type of experiment, seeding of cells should be done by either procedure 3.3.1 or 3.3.2. For field irradiation (e.g., for clonogenic survival) procedure 3.3.1 should be used. For immunofluorescence or experiments where coverslips are a requirement procedure 3.3.2 should be used.


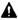
**CRITICAL STEP:** Culture should be near 90% confluence to yield sufficient number of irradiated cells.

#### 3.3.1. Seed The U2OS Cells at A Concentration of 100,000−200,000 Cells in A Mylar Dish in 2 mL of Medium. Culture Cells Overnight, Covering the Mylar Dishes Using the Lids of The LUMOX Dishes.


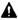
**CRITICAL STEP**: Growing other types of cells on Mylar could provide difficulty in attachment. Solutions to possible attachment problems are coating with laminin, gelatin or polylysine. In case of severe attachment problems, users could consider glow-discharged carbon coating [29].

#### 3.3.2. Seed U2OS Cells in a 6-wells Plate with Glass Coverslips (10 mm ø or smaller) in the Wells. Culture Cells for 24 h in the 6-wells Plate.


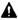
**CRITICAL STEP:** Be sure that the used coverslips are 10 mm ø or smaller.

### 3.4. Irradiation Procedure

Time of Completion: Depends on the required dose. For example: 2 Gray on a Mylar dish is 02:30 h and 2 Gray on a coverslip is 8 min. Workflow: The script is used to cover 19 positions under the Mylar dish for complete irradiation coverage of the dish. The timing of each position is entered in the program by the user and determines the dose deposited at that position. After each position has been irradiation (e.g., for 240.1 s) the collimator is redirected back to its original position and the irradiation is complete. Depending on the type of experiment, irradiation of cells should be done by either procedure 3.4.1. or 3.4.2. For field irradiation (e.g., for clonogenic survival) procedure 3.4.1. should be used. For immunofluorescence or experiments where cover slips are a requirement procedure 3.4.2. should be used. Figure 3 shows a schematic overview of the irradiation procedures.

#### 3.4.1. Irradiation of Cells Grown in the Mylar Dish

Turn on the computer and stage controller. Open MatLab script “Automated irradiation stage.m” in MatLab and press run.Turn on the heating in the irradiation box and wait for it to reach to 37 °C. Add 20 μL/mL HEPES (20 mm) buffer to each Mylar dish.Carefully place the Mylar dish in the dish holder. All Mylar dishes should have their lids on at all times.
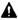
**CRITICAL STEP:** make sure the dish is level in the holder. A skewed positioning will affect the dose on the cells.Start the irradiation sequence in the controller software. Use Table 1 to calculate the irradiation time (in seconds) to fill in your time per irradiation point. Example: 1 Gy requires 240.1 s (4 min) of irradiation per position.Once the irradiation sequence is completed, carefully remove the Mylar dish from the irradiation setup. Depending on the type of sequential experiments, cells can now be trypsinized for further experimentation.
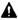
**CRITICAL STEP:** Cells have been in a variable temperature and non-buffered area for multiple h due to the irradiation time. This can affect the robustness of cells in general. Use control samples which undergo the same conditions as the experimental samples.

#### 3.4.2. Irradiation of Cells Grown on A Glass Cover Slip

Turn on the computer and stage controller. Open MatLab script “Automated irradiation stage.m” in MatLab and press run.Turn on the heating in the irradiation box and wait for it to reach to 37 °C.Remove the cover slip from the medium and wash with PBS. Try to remove as much as PBS as possible using paper tissues and put the cover slip up-side-down exactly in the center of the Mylar dish.
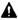
**CRITICAL STEP:** Be sure that the cells are facing down and are in between the Mylar and the glass cover slip.Carefully place the Mylar dish in the dish holder.Calculate your irradiation time using Table 1 and click on ‘Move to center’ in the MatLab irradiation window.Start timing using a stopwatch when the movement stage stops in the center and click on ‘Move to corner’ when the appropriate amount of minutes have passed to stop irradiation.Once the irradiation is completed, carefully remove the Mylar dish from the irradiation setup. Add 500 µL PBS in the Mylar dish to ‘lift’ the coverslip from the Mylar dish and carefully remove the coverslip.Return the coverslip back to the medium. The coverslip can now be used for further experimentation.

## 4. Expected Results

The described protocol has been validated using both clonogenic survival (whole dish irradiation) and super resolution microscopy (coverslip irradiation). By irradiating U2OS cells with alpha-particle and X-ray irradiation we compared the differences in cell survival and the differences in 53BP1 immunohistochemistry labeling.

### 4.1. Mylar Dish Irradiation for Clonogenic Survival

Using point-source irradiation allows in-depth analysis of DSBs induced by alpha particles while field-irradiation will allow experiments on larger number of cells for colony survival or immuno- blotting. A great difficulty for alpha particle irradiation is to assure that every cell has received the same dose. Our field-irradiation set up has been designed to possibly counter this problem. For validation of the Mylar dish irradiation a comparison of survival was made between X-ray or alpha particle irradiated samples. Alpha particle irradiated was done as described in this protocol. X-ray irradiation was done using the RS320 (Xstrahl Life Sciences, Surrey, United Kingdom), a self-contained cabinet, with a dose rate of 1.6554 Gy/min. Alpha particle irradiated cells showed severe decreased survival compared to X-ray irradiation cells (Figure 4). This severe effect confirms the effectiveness of the alpha particle irradiation protocol on cells growing on Mylar. In addition, assays showed low variation between three independent experiments.

### 4.2. Coverslip Irradiation for Super Resolution Microscopy

The imaging of cells affected by alpha particle irradiation can be done by conventional epi-fluorescent microscopes, but advanced confocal and super-resolution imaging techniques could provide more information. Due to the fact that DSBs induced by alpha particles could be present in multiple focus planes, we recommend acquiring 3D-images of affected cells to fully grasp the biological impact of alpha particles. Due to the possibility to use glass cover-slips, super resolution microscopy can be realized. To validate the irradiation of cells grown on glass coverslips, the DSB marker 53BP1 was used to investigate the differences between alpha particle and X-ray induced DSBs. After irradiation 53BP1 was marked using immunofluorescence and samples were imaged using SIM for detailed analysis (Figure 5). SIM imaging was performed on a Zeiss Elyra PS1 microscopy with an Andor iXon DU 885 EMCCD camera (Carl Zeiss AG, Oberkochen, Germany). The raw images were reconstructed into a high-resolution 3D-dataset using the Zeiss 2012 PS1 ZEN software. Reconstruction was done using default settings. 53BP1 foci showed similar structures in both alpha particle (Figure 5A) and X-ray irradiated samples (Figure 5B). In addition, quantification of the foci area revealed larger 53BP1 foci in alpha particle irradiated cells compared to X-ray irradiated cells (Figure 5C). These results show confirmation of cells irradiated by alpha-particles using the described protocol.

## Figures and Tables

**Figure 1 mps-02-00075-f001:**
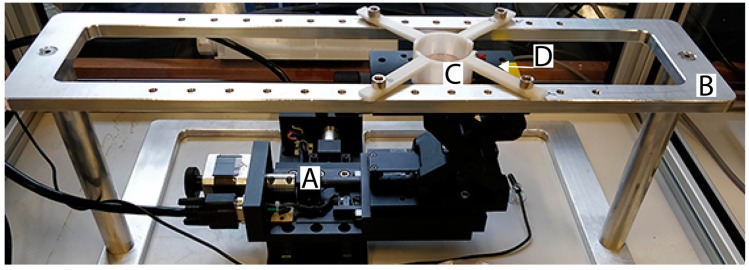
Overview of the automated external alpha particle irradiation set-up. (**A**) Two motorized linear stages connected to the Motion controller and the PC. (**B**) Aluminum frame as frame for the device (Appendix A). (**C**) Culture dish holder (Appendix A). (**D**) Radioactive source collimator (Appendix A).

**Figure 2 mps-02-00075-f002:**
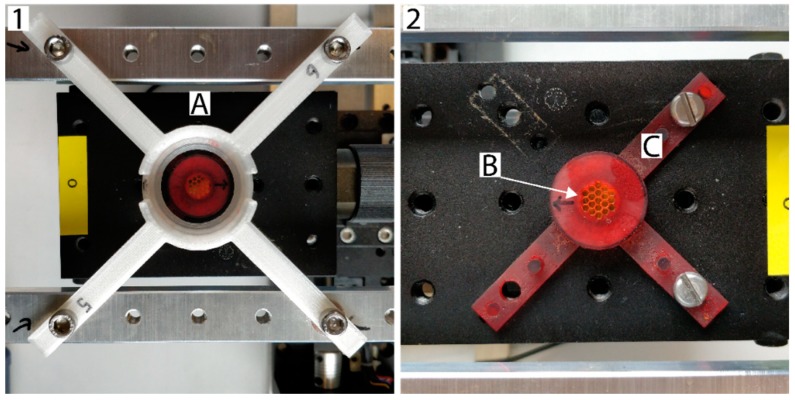
Top view of the irradiation set-up. Panel 1: (**A**) Top view with dish and source collimator in place. The source is aligned in the middle of the culture dish holder (Appendix A). Panel 2: (**B**) Radioactive surface protected by gold inside the source collimator. (**C**) 3-D printed source collimator (Appendix A).

**Figure 3 mps-02-00075-f003:**
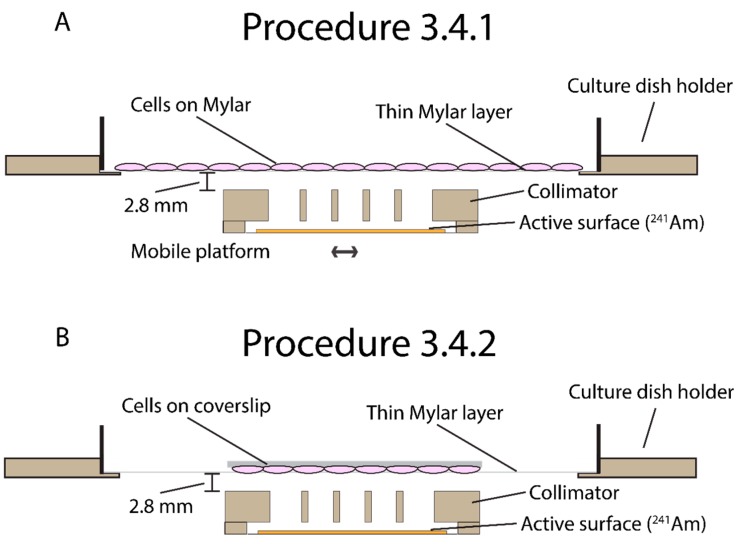
Schematic side views of both irradiation procedures. (**A**) Procedure 3.4.1 is meant for large field irradiation. Cells are grown in Mylar dishes. During the irradiation the active surface is mobile and irradiates the whole area covered with cells. (**B**). Procedure 3.4.2 is meant for coverslip irradiation. In this procedure the active surface is stationary and irradiates the coverslip.

**Figure 4 mps-02-00075-f004:**
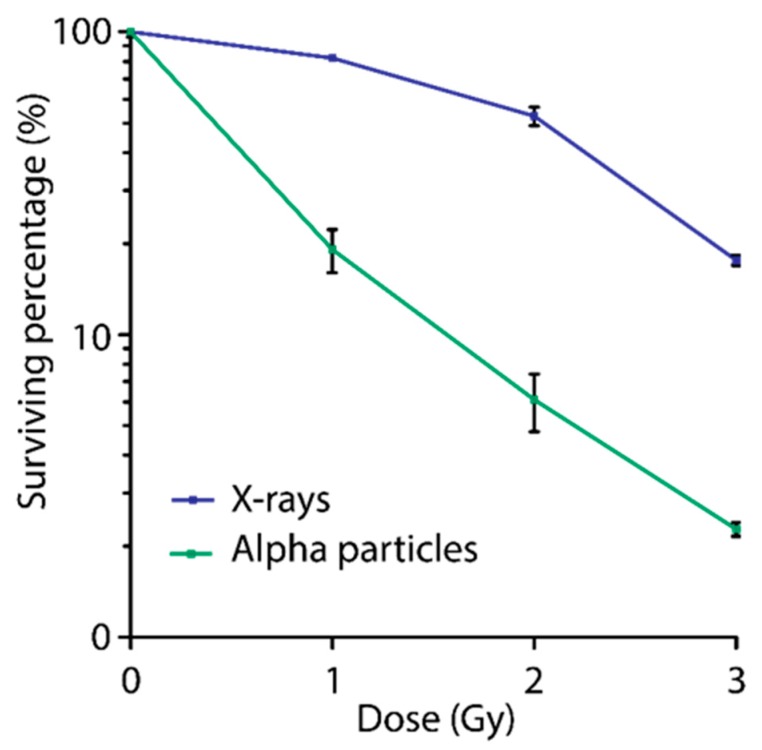
Clonogenic survival curve for U2OS cells irradiated using the alpha particle irradiation or X-rays. U2OS cells were seeded in Mylar dishes and treated with 1, 2 or 3 Gy of irradiation using both alpha particles or X-ray. Cells were trypsinized and seeded in triplicate. Colonies allowed to grow for 7 days (n = 3).

**Figure 5 mps-02-00075-f005:**
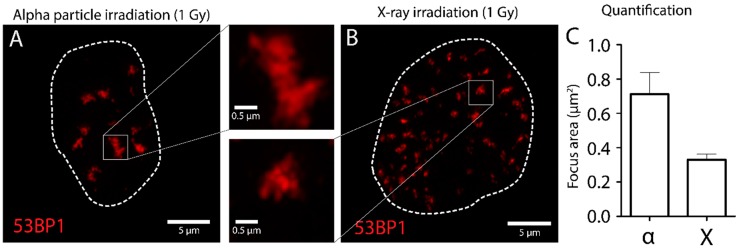
Nanoscopic analysis of DSBs in U2OS cells. U2OS are irradiated using external alpha particle irradiation (**A**) or X-ray (**B**), fixed after 1 h and stained for 53BP1 as DSB marker. SIM imaging was used for nanoscopic analysis of 53BP1 foci. Foci were quantified using ImageJ (**C**). Enlarged figures show 53BP1 foci in close up.

**Table 1 mps-02-00075-t001:** Characteristics of the described ^241^Am source and current setup. Adapted from [24]. Used for calculating the irradiation time.

Characteristic	Value
Area of active surface (mm^2^)	11
Source fluence at cell position (particles/s /cm^2^)	15966
Half-life (days)	157800
Distance between the active surface and mylar (mm)	5.0 (± 0.1)
Distance between collimator and Culture dish holder (mm)	2.8 (± 0.1)
Irradiation time per point (s/Gy) ^1^	240.1 (± 5.9%)

^1^ Calculated in an 8 µm layer of water above a 1.4 µm Mylar sheet, irradiated in 2017 using the described ^241^Am source.

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
