# Peer review of "Large Field Alpha Irradiation Setup for Radiobiological Experiments"

_mps, 2019, doi:10.3390/mps2030075_

Round 1
Reviewer 1 Report
General Comments
This is a useful description of techniques needed for irradiating cells with alpha particles. Some of the techniques described appear to be specific to a one-of-a-kind irradiator. It would be helpful generalize the protocol and identify which are general techniques that can be used with a variety of alpha particle irradiators and which are techniques that are specific to their irradiator. For example, lines 195-196 are important for all alpha irradiators and is a helpful warning.
There are numerous typos throughout. The document needs editing by native English speaker.
Specific Comments
p. 1, lines 58-60. “With the use of the right materials and conditions this protocol could yield fast and reliable answers to biological questions regarding alpha particle induced DSBs or cell survival after irradiation of different radionuclides.” Replace “different radionuclides” with “alpha particles”.
p. 1-2, lines 45-47. Commercially available sources from where? They are too small?
p. 2, lines 51-52. The statements regarding to nuclear track detectors should also refer to Gaillard et al. (1).
p. 2, line 55. Change “cross the Mylar-based …” to “avoid the Mylar-based”.
p. 6, line 203. What is a non-constant zone? Out of the incubator? Why hours?
p. 6, line 206. The table indicates that the distances tabulated are used to calculate the irradiation time? This is very puzzling. The fluence is not measured and then used to calculate the absorbed dose to an 8 um layer? This needs more explanation.
p. 6, line 223-224. There should be a check which excludes cover slips where the cells have grown outside the irradiation boundaries. Alternatively, there needs to be a way to keep track of cells that have not been irradiated.
p. 7, line 218. The cover slip will make the Mylar sag. How do you ensure that there isn’t a gap between the cover slip and Mylar?
p. 7, line 239 and Figure 4. It is very unlikely that all the cells receive the same absorbed dose when the mean dose is 0-3Gy. Some cells will be traversed more times than others and this is expected. Even so, a uniform irradiation of the Mylar surface should produce a monoexponential survival curve. The fact that it does not suggests that there may be issues with beam control as alluded to in the paragraph. Does the collimator produce dead spots?
p. 7. Fig. 4. Vertical axis should be “Surviving percentage (%)”
p. 7, line 241. It says that x-rays where used. What x-rays? How were they delivered? Filtration? Energy? Etc.
p. 8. Figure 5. The difference between the sizes of the foci is not convincing. The largest foci was selected in the case of alpha irradiation. There are plenty of foci in the same image that are the same size as ones found in the image for x irradiation. Maybe a distribution of foci sizes would be more convincing.
p. 10, line 286. References is not spelled correctly.
p. 10, lines 287-end. The authors appear to be unaware of the substantial body of literature on alpha particle irradiators. Citations to these articles should be in the manuscript
Supplement. The supplement has autocad files. Pdf’s of the same should also be included.
References that need to be cited.
(1-15)
1. Gaillard S, Pusset D, de Toledo SM, Fromm M, Azzam EI. Propagation distance of the alpha-particle-induced bystander effect: The role of nuclear traversal and gap junction communication. Radiat Res. 2009;171:513-520.
2. Inkret WC, Eisen Y, Harvey WF, Koehler AM, Raju MR. Radiobiology of alpha particles. I. Exposure system and dosimetry. Radiat Res. 1990;123:304-310.
3. Goodhead DT, Bance DA, Stretch A, Wilkinson RE. A versatile plutonium-238 irradiator for radiobiological studies with alpha-particles. Int J Radiat Biol. 1991;59:195-210.
4. Metting NF, Koehler AM, Nagasawa H, Nelson JM, Little JB. Design of a benchtop alpha particle irradiator. Health Phys. 1995;68:710-715.
5. Zarris G, Georgakilas AG, Sakelliou L, Sarigiannis K, Sideris EG. Alpha and gamma-irradiation of aqueous DNA solutions. Radiation Measurements. 1998;29:611-617.
6. Esposito G, Antonelli F, Belli M, et al. An alpha-particle irradiator for radiobiological research and its implementation for bystander effect studies. Radiation Research. 2009;172:632-642.
7. Seideman JH, Stancevic B, Rotolo JA, et al. Alpha particles induce apoptosis through the sphingomyelin pathway. Radiat Res. 2011;176:434-446.
8. M VJ, Shinde SG, S SK, et al. Dosimetry and radiobiological studies of automated alpha-particle irradiator. J Environ Pathol Toxicol Oncol. 2013;32:263-273.
9. Nilsson J, Bauden MP, Nilsson JM, Strand SE, Elgqvist J. Cancer cell radiobiological studies using in-house-developed alpha-particle irradiator. Cancer Biother Radiopharm. 2015;30:386-394.
10. Lee KM, Lee US, Kim EH. A practical alpha particle irradiator for studying internal alpha particle exposure. Appl Radiat Isot. 2016;115:304-311.
11. Soyland C, Hassfjell SP. A novel 210po-based alpha-particle irradiator for radiobiological experiments with retrospective alpha-particle hit per cell determination. Radiation and Environmental Biophysics. 2000;39:125-130.
12. Søyland C, Hassfjell SP, Steen HB. A new alpha-particle irradiator with absolute dosimetric determination. Radiat Res. 2000;153:9-15.
13. Neti PV, de Toledo SM, Perumal V, Azzam EI, Howell RW. A multi-port low-fluence alpha-particle irradiator: Fabrication, testing and benchmark radiobiological studies. Radiation Research. 2004;161:732-738.
14. Nawrocki T, Tritt TC, Neti P, Rosen AS, Dondapati AR, Howell RW. Design and testing of a microcontroller that enables alpha particle irradiators to deliver complex dose rate patterns. Phys Med Biol. 2018;63:245022.
15. Thompson JM, Elliott A, D'Abrantes S, Sawakuchi GO, Hill MA. Tracking down alpha-particles: The design, characterisation and testing of a shallow-angled alpha-particle irradiator. Radiat Prot Dosimetry. 2019;183:264-269.
Author Response
1, lines 58-60. “With the use of the right materials and conditions this protocol could yield fast and reliable answers to biological questions regarding alpha particle induced DSBs or cell survival after irradiation of different radionuclides.” Replace “different radionuclides” with “alpha particles”.Adjusted
1-2, lines 45-47. Commercially available sources from where? They are too small?
Commercially available sources from Eckert and Ziegler for example are too small in diameter too irradiate a full dish at once. Therefore needing an adjusted set-up. We have adjusted the text to make our message more clear
2, lines 51-52. The statements regarding to nuclear track detectors should also refer to Gaillard et al. (1).
We have added the reference
2, line 55. Change “cross the Mylar-based …” to “avoid the Mylar-based”.
Adjusted
6, line 203. What is a non-constant zone? Out of the incubator? Why hours?
We agree this sentence is confusing. The sentence has been adjusted to explain that cells in our set-up have been in an perplex box without proper temperature control, making the temperature of the cells not constant.
6, line 206. The table indicates that the distances tabulated are used to calculate the irradiation time? This is very puzzling. The fluence is not measured and then used to calculate the absorbed dose to an 8 um layer? This needs more explanation.
The table indicates the properties of current setup. The distances are measurements of how far the active surface is from the proposed position of the irradiated cells. The measurements of dose have been described before (see reference in title of table)
6, line 223-224. There should be a check which excludes cover slips where the cells have grown outside the irradiation boundaries. Alternatively, there needs to be a way to keep track of cells that have not been irradiated.
We agree with the reviewer on this; we have adjusted the protocol to where only coverslips should be used within the diameter of the radioactive surface. In addition, keeping track of cells which have not be irradiated during coverslip irradiation should be done via Immunohistochemistry of double strand break markers. Hereby the user excludes possible cells which have not been irradiated.
7, line 218. The cover slip will make the Mylar sag. How do you ensure that there isn’t a gap between the cover slip and Mylar?
To our knowledge is the possibility of a gap within the range of 0.1 mm negligible. This is due to that alpha particles attenuate minimally and the collimator divergence is small. We would also refer to Kouwenberg, J.J.M., et al., Alpha radiation dosimetry using Fluorescent Nuclear Track Detectors. Radiation Measurements, 2018. 113: p. 25-32 for further explaination.
7, line 239 and Figure 4. It is very unlikely that all the cells receive the same absorbed dose when the mean dose is 0-3Gy. Some cells will be traversed more times than others and this is expected. Even so, a uniform irradiation of the Mylar surface should produce a monoexponential survival curve. The fact that it does not suggests that there may be issues with beam control as alluded to in the paragraph. Does the collimator produce dead spots?
We do not fully agree that the survival curve is not monoexponentional. Validations has showed minor flaws within the 5% uncertainty. Indeed, the possibility of dead-spots could be present. Please refer to Kouwenberg, J.J.M., et al., Alpha radiation dosimetry using Fluorescent Nuclear Track Detectors. Radiation Measurements, 2018. 113: p. 25-32 for collimator description.
7. Fig. 4. Vertical axis should be “Surviving percentage (%)”
Adjusted
7, line 241. It says that x-rays where used. What x-rays? How were they delivered? Filtration? Energy? Etc.
Adjusted
8. Figure 5. The difference between the sizes of the foci is not convincing. The largest foci was selected in the case of alpha irradiation. There are plenty of foci in the same image that are the same size as ones found in the image for x irradiation. Maybe a distribution of foci sizes would be more convincing.
A quantification graph has been added
10, line 286. References is not spelled correctly.
Adjusted
10, lines 287-end. The authors appear to be unaware of the substantial body of literature on alpha particle irradiators. Citations to these articles should be in the manuscript
We thank the reviewer for making us aware on this literature we have added the references to the manuscript
Supplement. The supplement has autocad files. Pdf’s of the same should also be included.
Added
Reviewer 2 Report
Review for manuscript mps-554064 “Large field alpha irradiation setup for radiobiological experiments”.
The manuscript describes a large field alpha particle irradiator based on an isotopic source. The concept of using an isotopic alpha source to irradiate cells in culture is not a novel one and the authors should acknowledge prior work (for example Radiat. Res. 161, 732–738 (2004), Nucl Instr. Meth B 358, 26-31 (2015), Exp. Cell Res., 15, 529-536 (1958), but there are many more).
In my view the manuscript needs much more work both in terms of form and in terms of content, before it can be published.
Abstract:
Line 24 “In addition we present an adapted protocol that allows super resolution imaging”. Please rephrase as it seems to imply that you present a protocol for performing super resolution imaging Where in reality you just say that super resolution imaging was performed using Structured illumination Microscopy without providing any details (or even a reference describing the system you used).
Introduction:
The introduction seems disjointed and does not seem to give a coherent story on why this is important. The first two sentence refers to increased interest in the past year (quoting three papers from 2016) in isotope conjugated radiotherapy. In actuality the irradiator describes is just as applicable to radon risk estimation (which was a major interest to the radiobiology community in the 1990’s and 2000’s) and to alpha particle radiotherapy which was studied even earlier and may be returning to vogue. The introduction then describes LET/RBE of alpha particles and goes back to current studies
Page 1 line 39: “This effectiveness is due to the high energy… which causes short travel distance.” This sentence is not correct. The effectiveness is caused by the short distance between individual ionizations – on the order of 2 nm. See for example Fig 7.6 in Eric Hall’s book “Radiobiology for the Radiologist” or Figure 1c in Willers et al. Radiotherapy and Oncology 128 p68-75 (2018).
Experimental Design:
Page 3 line 98: please provide the intensity of the source (ideally also flux of alpha particles at cell location).
Page 3 line 111-112: please provide resolution/precision/repeatability of stages and jack.
Page 5 Line 154: Using acetone to bond the film to the ring is a nice idea and removes many of the issues associated with glue, which is commonly used for this purpose.
Page 6 Line 190: need to describe the script in more detail (a flow diagram would be good). The provided m file is missing the corresponding *.fig file and so cannot be run.
Page 6 Line 191: the irradiation box and heater are not described anywhere.
Page 6 line 198: The irradiation time depends on the intensity of the source which is not specified.
Page 6 Line 203: What is a “non-constant zone”? do you mean the exponential phase of cell growth?
Page 6 table 1: add source intensity and area. The caption says “using the described Am-241 source” but the source is not adequately described.
Page 7 Line 221: Why do you use a stopwatch? Can’t the script stop the irradiation more precisely?
Page 7 line 223: The area of the source should be specified.
Expected Results:
Are cells able to detach spontaneously from the membrane during or prior to irradiation? In such a case they will not be irradiated.
Page 7 line 239: “Assuring every cell has received the same dose”. Your system does NOT assure this. Cells receive a Poisson distribution of “doses” (actually particle traversals) around some mean which depends on the irradiation time and source intensity.
In order to assure that each cell has received the same dose you need a microbeam – a beam smaller than the cell where you can count particles traversing the cell, downstream of it. Using such a system you image each cell, position it above the beam and shut off the beam/move the cell when the required dose has been delivered. A simpler method, applicable in your setup, is to grow cells on track etch film and count particle tracks at the cell’s location (however that measures the cell’s dose rather than assures it is a specific value, see Chan et al, Radiation Measurements 43, pp S541-S545 (2008)).
Figure 4: How do you calculate dose delivered to the cells?
Section 5.2: SIM system is not described and no reference given.
Page 8 line 261: “Larger foci” –Please quantify.
Typos:
Please review manuscript carefully for consistency of font use. There are several places where the font changes in mid-sentence or between sentances. E.g.: Page 4 line 139, Page 7 line 234 vs line 237, Page 8 line 258.
Several places: “241-Americium” should probably be “241Am” or “Americium 241”.
Page 1 line 36: “upmost” should be “utmost”
Page 4 line 135: “3” should be “4”
Page 6 line 183: missing “I” in “Irradiation”
Page 6 line 184: Gray should not be plural
Page 8 line 250: “Triplets” should be “triplicate”
Author Response
The manuscript describes a large field alpha particle irradiator based on an isotopic source. The concept of using an isotopic alpha source to irradiate cells in culture is not a novel one and the authors should acknowledge prior work (for example Radiat. Res. 161, 732–738 (2004), Nucl Instr. Meth B 358, 26-31 (2015), Exp. Cell Res., 15, 529-536 (1958), but there are many more).
We agree, prior work has been acknowledged in the introduction in the update manuscript
In my view the manuscript needs much more work both in terms of form and in terms of content, before it can be published.
Abstract:
Line 24 “In addition we present an adapted protocol that allows super resolution imaging”. Please rephrase as it seems to imply that you present a protocol for performing super resolution imaging Where in reality you just say that super resolution imaging was performed using Structured illumination Microscopy without providing any details (or even a reference describing the system you used).
We agree with the reviewer and adjusted the sentence.
Introduction:
The introduction seems disjointed and does not seem to give a coherent story on why this is important. The first two sentence refers to increased interest in the past year (quoting three papers from 2016) in isotope conjugated radiotherapy. In actuality the irradiator describes is just as applicable to radon risk estimation (which was a major interest to the radiobiology community in the 1990’s and 2000’s) and to alpha particle radiotherapy which was studied even earlier and may be returning to vogue. The introduction then describes LET/RBE of alpha particles and goes back to current studies
We thank the reviewer for this comment; we have adjusted the introduction to a more coherent story.
Page 1 line 39: “This effectiveness is due to the high energy… which causes short travel distance.” This sentence is not correct. The effectiveness is caused by the short distance between individual ionizations – on the order of 2 nm. See for example Fig 7.6 in Eric Hall’s book “Radiobiology for the Radiologist” or Figure 1c in Willers et al. Radiotherapy and Oncology 128 p68-75 (2018).
We agree with the comment of the reviewer. The sentence has been adjusted.
Experimental Design:
Page 3 line 98: please provide the intensity of the source (ideally also flux of alpha particles at cell location).
We added the intensity of the source. We would like to refer to Kouwenberg, J.J.M., et al., Alpha radiation dosimetry using Fluorescent Nuclear Track Detectors. Radiation Measurements, 2018. 113: p. 25-32 for more information on the dosimetry of the source.
Page 3 line 111-112: please provide resolution/precision/repeatability of stages and jack.
This information is available on the distributors site using the Catalogues numbers
Page 5 Line 154: Using acetone to bond the film to the ring is a nice idea and removes many of the issues associated with glue, which is commonly used for this purpose.
We thank the reviewer for this compliment
Page 6 Line 190: need to describe the script in more detail (a flow diagram would be good). The provided m file is missing the corresponding *.fig file and so cannot be run.
The missing .fig file has been added. We agree with the reviewer the starting of the script should be explained more. This has been added to the manuscript.
Page 6 Line 191: the irradiation box and heater are not described anywhere.
Descriptions of these two items have been added
Page 6 line 198: The irradiation time depends on the intensity of the source which is not specified.
We added the intensity of the source. We would like to refer to Kouwenberg, J.J.M., et al., Alpha radiation dosimetry using Fluorescent Nuclear Track Detectors. Radiation Measurements, 2018. 113: p. 25-32 for more information on the dosimetry of the source.
Page 6 Line 203: What is a “non-constant zone”? do you mean the exponential phase of cell growth?
We agree this sentence is confusing. The sentence has been adjusted to explain that cells in our set-up have been in an perplex box without proper temperature control, making the temperature of the cells not constant.
Page 6 table 1: add source intensity and area. The caption says “using the described Am-241 source” but the source is not adequately described.
We added the intensity of the source. We would like to refer to Kouwenberg, J.J.M., et al., Alpha radiation dosimetry using Fluorescent Nuclear Track Detectors. Radiation Measurements, 2018. 113: p. 25-32 for more information on the dosimetry of the source.
Page 7 Line 221: Why do you use a stopwatch? Can’t the script stop the irradiation more precisely?
We agree the script could do this more precisely. However, the script is based on the irradiation of the while Mylar Dish. When we use Procedure 3.4.2 only a small part of the Mylar dish should be irradiated. For now we have chosen to use a stopwatch to time the irradiation time and move the source back to its original position.
Page 7 line 223: The area of the source should be specified.
The active area of the source has been added.
Expected Results:
Are cells able to detach spontaneously from the membrane during or prior to irradiation? In such a case they will not be irradiated.
We believe that indeed during or prior the irradiation cells could detach from the Mylar. In this case these cells will not be (properly) irradiation. However, since the protocol suggests to wash after the irradiation step only cells are used in the assays which were properly attached and irradiated.
Page 7 line 239: “Assuring every cell has received the same dose”. Your system does NOT assure this. Cells receive a Poisson distribution of “doses” (actually particle traversals) around some mean which depends on the irradiation time and source intensity.
In order to assure that each cell has received the same dose you need a microbeam – a beam smaller than the cell where you can count particles traversing the cell, downstream of it. Using such a system you image each cell, position it above the beam and shut off the beam/move the cell when the required dose has been delivered. A simpler method, applicable in your setup, is to grow cells on track etch film and count particle tracks at the cell’s location (however that measures the cell’s dose rather than assures it is a specific value, see Chan et al, Radiation Measurements 43, pp S541-S545 (2008)).
We completely agree with the reviewer. Perhaps the sentence is misleading; we do not claim that our system does assure this. We state that this has been a difficult problem in the field. We are very intrigued by the reviewers suggestions using track etch film or FNTDs to grow cells and possibly image the traversals per cell. However, we suspect to encounter that no alpha particle achieve to reach the film/FNTD after dose delivery in a whole cell.
Figure 4: How do you calculate dose delivered to the cells?
The dose delivery to cells has been adapted from previous dosimetry work Kouwenberg, J.J.M., et al., Alpha radiation dosimetry using Fluorescent Nuclear Track Detectors. Radiation Measurements, 2018. 113: p. 25-32
Section 5.2: SIM system is not described and no reference given.
Details of the SIM system have been added.
Page 8 line 261: “Larger foci” –Please quantify.
A quantification graph has been added
Typos:
Please review manuscript carefully for consistency of font use. There are several places where the font changes in mid-sentence or between sentences. E.g.: Page 4 line 139, Page 7 line 234 vs line 237, Page 8 line 258.
We apology to the reviewer for this inconsistency. We adjusted the font use and typos throughout the manuscript.
Reviewer 3 Report
As there are very few alpha irradiation facilities in the world, the author should list at least some of the some in the Introduction and tie the references to the facilities where these references have been generated.
Just one cell line as a experimental proof is not sufficient, the data (at least the survival data) for minimun of two cell lines should be presented.
Author Response
As there are very few alpha irradiation facilities in the world, the author should list at least some of the some in the Introduction and tie the references to the facilities where these references have been generated.
We thank the reviewer for this suggestion. Several references have been added to the manuscript.
Just one cell line as a experimental proof is not sufficient, the data (at least the survival data) for minimun of two cell lines should be presented.
As we mention in the protocol, adaptation to the cell line of choice requires adjustments to the culture conditions. We want to focus on the protocol of irradiation and not the culture condition itself we chose 1 cell line to represent expected results and not proof of difference in survival. We hope we can convince the reviewer that adjusting the culture conditions to other cell lines is another goal and not relevant for this protocol. If the reviewer suggests we propose tips to adjust culture conditions to the user needs we could provide this.
Round 2
Reviewer 2 Report
The authors have adequately addressed most of my comments. Below are a few additional ones, based on the revised manuscript:
Line 32: "components" is probably not the right word here.
Line 36: pls. add comma after "particles"
Line 47: add "from" before "Eckert"
Line 66:"developed for the need" is not gramatically correct
Line 68: "build" should be "built"
Line 108: are the cover slips really <1cm in diameter? searching for the part number on Fisher's web site does not return anything.
Line 112,113: The resolution/repeatability is important for the operation of the system and should be given.
Line 117: What is the material of the collimator?
Line 118: Perspex is a trade name. The actual material is Acrylic or Poly-Methyl Methacrylate (PMMA). What is the volume of the box?
Figure 2: please label the panels.
Line 184: "irradiates only part of the coverlsip" how can you tell which cells were irradiated? The response to one of the other reviewers that you would use 53BP1 foci to identify the irradiated cells is not a good one. If you are studying 53BP1 then scoring only positive cells would be a major source of errors. If you are not studying these foci then you need an additional analysis step which may interfere with the primary analysis and you still have the possibility of not scoring Foci-less irradiated cells or unirradiated cells with spontaneous foci.
Line 196:Please describe the workflow of the script.
Line 204:When irradiating multiple spots on a mylar dish is there an overlap or a gap between irradiation fields? How is this verified?
Line 210: Cells would likely be happier if humidified warm air is blown on them, rather than using a hair dryer, which, by definition, blows dry air.
Table 1: please add half life of 241Am. area is given in units of length. please correct
Line 251: "live" should be "life"
Line 302: there is a weird abbreviation.
Author Response
The authors have adequately addressed most of my comments. Below are a few additional ones, based on the revised manuscript:
We adjusted all textual comments and possible typos as reviewer pointed out.
Line 108: are the cover slips really <1cm in diameter? searching for the part number on Fisher's web site does not return anything.
We would like the refer to the following link. There are some coverslips which reach down to 5 mm in diameter.
https://www.thermofisher.com/order/catalog/product/CB00100RA020MNT0?SID=srch-srp-CB00100RA020MNT0
Line 112,113: The resolution/repeatability is important for the operation of the system and should be given.
Added
Line 117: What is the material of the collimator?
Added
Line 118: Perspex is a trade name. The actual material is Acrylic or Poly-Methyl Methacrylate (PMMA). What is the volume of the box?
Added and adjusted
Figure 2: please label the panels.
Adjusted figure 2
Line 184: "irradiates only part of the coverlsip" how can you tell which cells were irradiated? The response to one of the other reviewers that you would use 53BP1 foci to identify the irradiated cells is not a good one. If you are studying 53BP1 then scoring only positive cells would be a major source of errors. If you are not studying these foci then you need an additional analysis step which may interfere with the primary analysis and you still have the possibility of not scoring Foci-less irradiated cells or unirradiated cells with spontaneous foci.
We agree with the reviewer. As we have adjusted the manuscript to use only coverslips which are in equal diameter or smaller as the radioactive surface we assume that each cell has been irradiated. We would again like to refer to FNTD-images of Kouwenberg et al (2016); we see a very condense image of alpha particles which have been absorbed in the FNTD crystal. Of course, the probability of particles actually hit cells is dependent on seeding density and the used dose. In addition, when larger coverslips are used the user is free to start the irradiation sequence to cover the whole glass.
Line 196:Please describe the workflow of the script.
We do not fully understand the reviewer on the workflow of the script. We have added a description in the text on how the script passes the 19 positions which are programmed for the complete coverage of the dish. Each position will then be addressed for a certain amount of time (which should be introduced by the user). This time will then determine the dose deposited. We hope that this brief explanation is sufficient in the manuscript. Otherwise, we would like to refer to the adjusted original matlab script.
Line 204:When irradiating multiple spots on a mylar dish is there an overlap or a gap between irradiation fields? How is this verified?
The honeycomb design of the collimator has been verified for each position in the MatLab script. As described in Kouwenberg et al (2016) the positions 1-19 seamlessly connect to each other to create a virtual larger irradiation area. Theoretically there is no overlap; however as seen on radio chromic films there could be minor overlap which is neglect able (max 5%).
Line 210: Cells would likely be happier if humidified warm air is blown on them, rather than using a hair dryer, which, by definition, blows dry air.
We agree with the reviewer. In terms of adjusting the set-up for more biological sound experiments, it can be still improved
Table 1: please add half-life of 241Am. area is given in units of length. please correct
Added and corrected
Line 302: there is a weird abbreviation.
This is a reference. The abbreviation is in the name of this Task Group.
Reviewer 3 Report
The manuscript is acceptable for publication after the revision, however,
it is a reviewer's recommendation that authors provide a supplementary file with the advice to potential users on adjustment of their cell culture conditions.
Author Response
Reviewer 3 (2e ronde)
The manuscript is acceptable for publication after the revision, however,
it is a reviewer's recommendation that authors provide a supplementary file with the advice to potential users on adjustment of their cell culture conditions.
We agree with the reviewer and added an CRITICAL STEP after the step where the user seeds the cells. Here we suggest possible solutions to attachment problems on mylar.